# Solid-State Fermentation as Green Technology to Improve the Use of Plant Feedstuffs as Ingredients in Diets for European Sea Bass (*Dicentrarchus labrax*) Juveniles

**DOI:** 10.3390/ani13172692

**Published:** 2023-08-23

**Authors:** Lúcia Vieira, Diogo Filipe, Diogo Amaral, Rui Magalhães, Nicole Martins, Marta Ferreira, Rodrigo Ozorio, José Salgado, Isabel Belo, Aires Oliva-Teles, Helena Peres

**Affiliations:** 1Department of Biology, Faculty of Sciences, University of Porto, 4169-007 Porto, Portugal; 2Interdisciplinary Centre of Marine and Environmental Research (CIIMAR-UP), 4050-123 Porto, Portugal; 3Centre of Biological Engineering, University of Minho, 4710-057 Braga, Portugal; 4Industrial Biotechnology and Environmental Engineering Group “BiotecnIA”, Chemical Engineering Department, University of Vigo, Campus Ourense, As Lagoas s/n, 32004 Ourense, Spain; 5LABBELS—Associate Laboratory in Biotechnology and Bioengineering and Microelectromechanical Systems, 4710-057 Braga, Portugal; 6LABBELS—Associate Laboratory in Biotechnology and Bioengineering and Microelectromechanical Systems, 4704-553 Guimarães, Portugal

**Keywords:** aquaculture, exogenous enzymes, solid-state fermentation, plant feedstuffs, technological process

## Abstract

**Simple Summary:**

The rapid growth of the world’s population has increased the demand for seafood, leading to the expansion of aquaculture to fulfill these needs and reduce the pressure on wild fish stocks. Plant feedstuffs (PFs) are often used as the main protein source in aquafeeds due to their wide availability and low cost. However, PFs usually contain high levels of non-starch polysaccharides that limit their utilization in aquafeeds, mainly for carnivorous fish. Solid-state fermentation (SSF) is a cost-effective technological process that may reduce anti-nutritional factor levels while improving nutrient digestibility and the production of several bioactive compounds, enhancing feedstuffs’ nutritional value in aquafeeds. Hence, this study evaluated the effects of using a PF mixture (rapeseed, soybean, rice bran, and sunflower seed meals, 25% each) solid-state fermented (SSFed) with *Aspergillus niger* CECT 2088 at two inclusion levels (20% and 40%) on European sea bass juveniles’ growth performance, feed digestibility, digestive and catabolic enzyme activity, and plasma metabolites. Overall, the SSFed PF mixture improved the overall feed digestibility, and utilization efficiency, when included at balanced level (20%) without negatively impacting fish growth performance, but not at the higher level (40%).

**Abstract:**

This study aimed to evaluate the utilization by juvenile European sea bass of a SSFed PF mixture with *Aspergillus niger* CECT 2088. A 22-day digestibility and a 50-day growth trial were performed testing four diets, including 20 or 40% of an unfermented or SSFed PF mixture (rapeseed, soybean, rice bran, and sunflower seed meals, 25% each). SSF of the PF added cellulase and β-glucosidase activity to the diets. Mycotoxin contamination was not detected in any of the experimental diets except for residual levels of zearalenone and deoxynivalenol (100 and 600 times lower than that established by the European Commission Recommendation-2006/576/EC). In diets including 20% PF, SSF did not affect growth but increased apparent digestibility coefficients of protein and energy, feed efficiency, and protein efficiency ratio. On the contrary, in diets including 40% PF, SSF decreased growth performance, feed intake, feed and protein efficiency, and diet digestibility. SSF decreased the intestinal amylase activity in the 40% SSFed diet, while total alkaline proteases decreased in the 20% and 40% SSFed diets. Hepatic amino acid catabolic enzyme activity was not modulated by SSF, and plasma total protein, cholesterol, and triglyceride levels were similar among dietary treatments. In conclusion, dietary inclusion of moderate levels of the SSFed PF, up to 20%, improves the overall feed utilization efficiency without negatively impacting European sea bass growth performance. The replacement of PF with the SSFed PF mixture may contribute to reducing the environmental footprint of aquaculture production.

## 1. Introduction

European sea bass (*Dicentrarchus labrax*) is an important aquaculture species in the Mediterranean, accounting for 2.9% of the global production of marine and coastal aquaculture fish, with 243.9 thousand tonnes in 2020, of which 81.4 thousand tonnes were produced in Europe [1]. As a carnivorous fish, protein makes up a large part of its diet [2] and is also the most expensive component of aquaculture [3].

Fishmeal has traditionally been used as the main protein source in carnivorous fish diets due to its high digestibility, palatability, and adequate amino acid and fatty acid profiles [4]. However, it is an expensive protein source due to its low availability [5]. New alternative protein sources have been studied, including insect meals and fungal, algal, and processed animal proteins [6]. However, plant feedstuffs (PFs) are still the main alternative protein source due to their wide availability, relatively low cost, and higher environmental friendliness than fishmeal [6,7]. Nevertheless, PFs have low palatability, unbalanced amino acid profiles, and contain antinutritional factors (ANFs), such as non-starch polysaccharides (NSPs), which limit their use [8,9] and increase waste production [10]. Therefore, the efficient use of these ingredients by fish is of great interest. Some strategies can be applied to improve PF utilization, such as supplementing the feed with flavorings [11], amino acids [12], or exogenous enzymes [13], or using pre-treatments to improve the nutritional value of the ingredient and reduce ANF. Different pre-treatments are used to remove ANF, such as soaking, milling, debranding, roasting, cooking, sprouting, and fermentation [14].

Solid-state fermentation (SSF) is a process that increases the nutritional value of PF [15,16,17]. SSF is mainly characterized by using low free water [18] and microorganisms, such as filamentous fungi, which efficiently penetrate the substrate through their hyphal growth [19].

Fungi produce several valuable bioactive compounds such as enzymes, organic acids, antibiotics, pigments, and antioxidants [18,20], which have myriad industrial applications, especially as additives in the feed industry [21]. Among them, *Aspergillus niger*, a fungus generally regarded as safe by the Food and Drug Administration (FDA), is known for producing various enzymes like phytases, lipases, proteases, and carbohydrases such as cellulases and xylanases [22,23,24]. Since the conditions used in SSF are very similar to the natural habitat of these fungi, the growth of the microorganism can alter the composition of the substrate on which it grows. In particular, fungi are described as enriching lignocellulosic materials with microbial proteins and enzymes. In this way, crude fiber content is reduced, and crude protein, protein solubility, and protein and fiber digestibility are increased [25,26], increasing the nutritional value of PF for aquaculture use.

There are already some reports describing the benefits of SSF for growth performance and feed efficiency [27,28,29], feed digestibility [30], feed palatability [31], innate immunity [32], and stress tolerance [33] in fish. To date, there is limited information on the effects of PF treated using SSF with *A. niger* on fish growth performance and feed digestibility.

The current study aimed to investigate the effects of dietary supplementation of solid-state fermented (SSFed) PF mixture with *A. niger* CECT 2088 on growth performance, feed digestibility, plasma metabolites, gut digestive enzyme, and liver catabolic enzyme activities of European sea bass juveniles.

## 2. Materials and Methods

This study was approved by the ORBEA Animal Welfare Committee of CIIMAR (ORBEA; reference ORBEA_CIIMAR_27_2019) according to the European Union directive 2010/63/EU and the Portuguese Law DL 113/2013. All procedures were performed by certified scientists (Functions a, b, c and d defined in article 23 of European Union Directive 2010/63), in compliance with the Federation of Laboratory Animal Science Association (FELASA, London, UK) recommendations and the EU Directive 2010/63/EU on the protection of animals for scientific purposes.

### 2.1. Solid-State Fermentation

A PF mixture of rapeseed, soybean, rice bran, and sunflower seed meals, 25% each, reflecting the average dietary incorporation of these ingredients in practical diets, except for rice bran, was prepared. This mixture was used as a substrate for SSF with *Aspergillus niger* CECT 2088 (Spanish collection of type cultivars, Paterna, Spain) to obtain modified biomass enriched with mycoprotein, improved protein digestibility, and reduced dietary fiber due to the production of carbohydrases.

*Aspergillus niger* CECT 2088, preserved at −80 °C in glycerol, was cultivated on malt extract agar (MEA) slants, incubated for 7 days at 25 °C, and then stored at 4 °C. Spores were then suspended and their concentration used in the SSF adjusted with a Neubauer counting chamber. Before the SSF, the PF mixture was sterilized using an autoclave (121 °C for 15 min). Then, SSF was performed with 400 g of substrate inoculated with 80 mL of spore suspension containing 10^6^ spores mL^−1^ and maintained at 25 °C and 75% humidity (*w*/*w*) on a wet basis in tray bioreactors (43 × 33 × 7 cm) for 7 days.

The substrate was stirred daily to ensure proper substrate oxygenation due to the 5 cm height of the substrate bed, and the tray was covered with plastic film with small holes. After SSF, the fermented mixture was dried at 60 °C for 3 days [32].

### 2.2. Experimental Diets

Four isoproteic and isolipidic diets (42% crude protein and 18% crude lipids) were formulated containing the non-fermented and fermented PF mixes at 20% (20 Mix and 20 SSF) and 40% (40 Mix and 40 SSF). Dietary ingredients were ground, mixed, and pelleted through a 3 mm thick diameter using a laboratory pellet mill (California Pellet Mill, Crawfordsville, IN, USA). After pelleting, dietary ingredients were dried (60 °C, 48 h) and stored at −20 °C until use. The ingredients and proximate composition of the experimental diets are listed in Table 1.

### 2.3. Digestibility Trial

This trial was conducted in a RAS comprising 12 fiberglass tanks of 60 L capacity equipped with fecal sedimentation columns [34] connected to the outlet of each tank. Five European seabass juveniles (initial body weight: 70.9 g) were distributed to each tank, the chromic oxide was added as a digestibility marker (0.5% DM), and the diets were randomly assigned to triplicate groups. 

Fish were acclimated to the diets and experimental conditions for 7 days; then, feces were collected daily for 14 days. Fish were hand-fed twice daily until satiation, and before the first daily meal, feces were collected from the sedimentation column, centrifuged (4000× *g*; 10 min), and pooled frozen for each tank. Approximately 30 min after the last daily meal, the tanks, pipes, and settling columns were thoroughly cleaned to remove residual feces and uneaten feed. During the trial, the water temperature was set at 24 °C, water flow at 5 L·min^−1^, salinity at 35‰, and photoperiod at 12 h light:12 h dark.

Apparent digestibility coefficients (ADCs) of dry matter, protein, lipids, and gross energy of the experimental diets were calculated as follows:


(1)
$${ADC}_{diet}=\left(1-\left(\frac{dietary\;{Cr}_{2}O_{3}\;level\times feces\;nutrient\;or\;energy\;level}{feces\;nutrient\;or\;energy\;level\times dietary\;{Cr}_{2}O_{3}\;level}\right)\right)\times 100$$


### 2.4. Growth Trial 

The growth trial was conducted at the Interdisciplinary Centre of Marine and Environmental Research (CIIMAR), University of Porto, Portugal, using European sea bass (*Dicentrarchus labrax*) juveniles purchased from a certified hatchery (Sonríonansa S.L., Cantabria, Spain). Fish were transported to the experimental facilities, quarantined for 15 days in a recirculating aquaculture system (RAS) with a 2000 L tank, and fed a commercial diet (NEO diet, Aquasoja; Sorgal, S.A., Ovar, Portugal).

Fish were then transferred to the experimental system, which consisted of a thermo-regulated RAS with 12 tanks of 300 L capacity. Fish were acclimatized to the experimental system for 2 months and fed a commercial diet (NEO diet, Aquasoja; Sorgal, S.A., Ovar, Portugal). At the beginning of the growth trial, 12 homogeneous groups of 20 fish (initial body weight of circa 10 g) were formed. The experimental diets were randomly allocated to triplicate groups, and during the trial, fish were hand-fed twice a day for six days a week until apparent satiation. The trial lasted for 50 days. Dead fish weight was recorded.

During the trial, a water flow of 5 L min^−1^ was maintained in each tank. Water temperature (24 ± 1 °C), salinity (35 ± 1‰), dissolved oxygen (7 ± 1 mg L^−1^), and nitrogenous compounds (<0.02 mg L^−1^) were recorded daily. Photoperiod was kept at 12 h light: 12 dark.

### 2.5. Sampling

At the start of the growth trial, five fish were sampled from the initial population for whole-body composition analyses. At the end of the growth trial, fish were bulk-weighed following one day of feed deprivation to determine the growth parameters. Then, three fish from each tank were randomly sampled and sacrificed with anesthetic overdose (0.6 mL L^−1^) to determine whole-body composition [35]. Another three fish were randomly sampled 4 h after the first meal at the end of the trial. Blood was collected from the caudal vein with heparinized syringes, centrifuged at 11,200× *g* for 10 min, and the plasma was stored at −80 °C until analysis. Then, fish were sacrificed using cervical dislocation followed by decapitation, and the liver, intestine, and muscle were collected and stored at −80 °C for posterior analysis. 

### 2.6. Carbohydrase Activity 

For the determination of enzyme activity, an aqueous extraction of the SSF substrate was performed. Briefly, water was added to the substrate at a 1:5 ratio (1 g solute dry weight/5 mL solvent) and left with constant stirring for 30 min. The substrate was then filtered through a fine-mesh net, centrifuged at 11,200× *g* for 10 min at 4 °C, and the supernatant was vacuum filtered through filter paper (11 µm pore size). The resulting extract was stored at −20 °C until analysis.

Cellulase (endo-1,4-β-glucanase) and xylanase (endo-1,4-β-xylanase) activities were indirectly determined using the release of reducing sugars during incubation of the extracts with carboxymethylcellulose (2%) or xylan (1%) at 50 °C for 30 and 15 min, respectively. Reducing sugars were quantified according to the DNS method described by Fernandes et al. [36]. One unit of enzyme activity (U) was defined as the amount of enzyme required to release 1 µmol of glucose for cellulase or xylose for xylanase per minute under assay conditions.

The activity of β-glucosidase was determined using 5 mM 4-nitrophenyl β-d-glucopyranoside (pNPG) as the substrate, according to Sousa et al. [16]. 4-Nitrophenol was used as a standard, and one unit (U) of enzymatic activity was defined as the amount of enzyme required to release 1 μmol of glucose or p-nitrophenol per minute. Results were expressed in units per gram (U g^−1^). 

### 2.7. Antioxidant Determination

To determine the antioxidant activity of the diets, a double extraction was performed with water/methanol (50:50 ratio) and with acetone/water (70:30 ratio) for 30 min under constant stirring at room temperature. Subsequently, samples were filtered and centrifuged as described for enzyme determination. 

Total antioxidant activity was determined using the DPPH (2,2-diphenyl-1-picrylhydrazyl) method described by Dulf et al. [37] with modifications [16]. Trolox (6-hydroxy-2,5,7,8-tetramethylchroman-2-carboxylic acid) was used as a standard, and the antioxidant activity of the extracts was expressed as μmol of Trolox equivalents per gram of dry substrate.

Total phenolic compounds in the extracts were accessed as described in Filipe et al. [38]. Gallic acid was used as a standard, and phenolic activity was expressed as mg gallic acid equivalents (GAEs) per gram of sample.

### 2.8. Proximate Analysis

The chemical composition of the ingredient mixture, SSF products, diets, feces, and whole fish was determined following the standard procedures of the Association of Official Analytical Chemists [39]. Briefly, dry matter (DM) was determined by drying overnight or until a constant weight at 105 °C and ash by incineration in a muffle furnace at 450 °C for 16 h. Crude protein (N×6.25) was analyzed via the Kjeldahl method using a Kjeltec digestion and distillation unit (Tecator Systems, Höganäs, Sweden; models 1015 and 1026, respectively). Crude lipids were determined via extraction with petroleum ether using a Soxtec HT System (Tecator, Höganäs, Sweden). Gross energy was analyzed via combustion in an adiabatic bomb calorimeter (PARR Instruments, Moline, IL, USA; PARR model 1261). Chromium oxide content in the diets and feces was quantified using acid digestion, according to Bolin et al. [40].

### 2.9. Major Mycotoxins’ Determination

The experimental diets were subjected to mycotoxin analysis using a validated multi-metabolite (>800) liquid chromatography/electrospray ionization–tandem mass spectrometric (LC/ESI–MS/MS) method, as described previously by Steiner et al. [41] and Sulyok et al. [42]. Briefly, milled samples of 5 g were placed into 250 mL Erlenmeyer flasks with 20 mL of extraction solvent (acetonitrile/water/acetic acid 79:20:1, *v*/*v*/*v*). After agitation with a rotary shaker for 90 min, the solvent solution–sample mixture was centrifuged for 2 min at 2012× *g*. The extract was diluted with dilution solvent (1:1). Identification and quantification of each analyte were performed in the mode of multiple reaction monitoring with positive and negative polarity in two separate chromatographic runs using a QTrap 5500 LC-MS/MS system (Applied Biosystems, Foster City, CA, USA) equipped with a TurboV electrospray ionization (ESI) source coupled to a 1290 series UHPLC system (Agilent Technologies, Waldbronn, Germany). 

### 2.10. Digestive and Intermediary Metabolism Enzyme Activity Determination 

Intestine and liver samples were homogenized (1:6 *w*/*v*) in ice-cold buffer (100 mM Tris-HCl, 0.1% mM EDTA, and 0.1% (*v*/*v*) Triton X-100, pH 7.8). Samples were then centrifuged at 30,000× *g* for 30 min at 4 °C, and the supernatants were stored at −80 °C until analysis. Samples were measured at 37 °C in a Multiskan GO microplate reader (Model 5111 9200; Thermo Scientific, Nanjing, China). Intestine samples were used to determine the activity of digestive enzymes, while the activity of enzymes of intermediary metabolism was measured in the liver. 

α-Amylase (EC 3.2.1.1) and lipase (EC 3.1.1.3) were determined using commercial kits (Spinreact ref. 1001274 and 41201, respectively) adapted to European sea bass, as described by Magalhães et al. [43]. Total alkaline proteases were determined according to Fernandes et al. [30].

Liver aspartate (ASAT/GOT, EC 2.6.1.1) and alanine aminotransferase (ALAT/GPT, EC 2.6.1.2) activities were determined using commercial kits (Spinreact, ASAT/GOT; 41,273; ALAT/GPT; 41,283). Glutamate dehydrogenase (GDH, EC 1.4.1.2) activity was calculated according to Morales et al. [44].

Enzymatic activity was expressed in units (α-amylase, lipase, protease) or milliunits (ALAT, ASAT, and GDH). One unit (U) of enzymatic activity was described as the amount of enzyme required to release 1 μmol of substrate per min. All enzyme activities were expressed per mg of soluble protein. Soluble protein was calculated according to Bradford [45] with a protein assay kit (Biorad, Ref. 5000006; Algés, Portugal) using bovine serum albumin as the standard.

### 2.11. Plasma Metabolites’ Determination

Analyses of plasma glucose, cholesterol, triglycerides, and total protein were performed using commercial kits from SpinReact, S.A. (Gerona, Spain; ref.: 1001191, ref.: 1001091, ref.: 41031, and ref.: 1001291, respectively). All plasmatic parameters were analyzed in a Multiskan GO microplate reader (Model 5111 9200; Thermo Scientific, Nanjing, China).

### 2.12. Statistical Analysis 

All data were checked for normality and homogeneity of variances and normalized when applicable. Data were analyzed using two-way ANOVA with SSF and the inclusion level as independent variables. If an interaction was significant, a *t*-test was performed for each factor. *p* < 0.05 was used as the significance level for all statistical analyses. Statistical analyses were performed with IBM SPSS Statistics software version 26 (IBM, New York, NY, USA).

## 3. Results

Cellulase and β-glucosidase activities were detected in the SSF diets, while no xylanase activity was detected (Table 2). Total antioxidant levels were reduced in the SSF diets, while phenolic levels were unaffected by dietary treatments.

The mycotoxins, aflatoxin B1, T-2 toxin, fumonisin, and ochratoxin A were not detected in any experimental diet. Zearalenone, deoxynivalenol, and the sum of ergot alkaloids were detected in the 20 SSF diet, and zearalenone and the sum of ergot alkaloids in the 40 SSF diet (Table 2).

The apparent digestion coefficients (ADCs) of the diets were not affected by the PF inclusion level but by SSF treatment (Table 3). The ADCs of protein, energy, and dry matter were higher in the 20 SSF diet than in the 20 Mix diet, while the opposite was observed in the 40 Mix groups.

No signs of disease were observed during the growth trial. Mortality was low but higher in the SSF groups than in the non-SSF groups (Table 4). Feed intake (FI) was affected by both the PF inclusion level and SSF, being lower in 40% and SSF groups. Growth performance was also influenced by diet composition and was lower in fish fed the 40 Mix diets. Within the 40 Mix diets, growth was also significantly lower in fish fed the SSF diet. Feed efficiency (FE) was higher in fish fed the 20 SSF diet than the fish 20 Mix diet, while the opposite was observed in the 40 Mix groups. A similar pattern was observed for protein efficiency ratio (PER) and nitrogen retention (NR).

Whole-body ash and lipid content were unaffected by diet. Dry matter content was higher in fish fed the 20 Mix than the 40 Mix diet and, within the 20% groups, it was higher in fish fed the 20 Mix than the 20 SSF diet. Protein content was also higher in fish fed the 20 Mix than the 20 SSF, and energy content was higher in fish fed the 20% than the 40% (Table 5). 

Intestinal amylase and lipase activity was higher in fish fed the 40 Mix diet than in fish fed the 20 Mix diet. SSF increased lipase activity in the 20% group, while amylase activity decreased in the 40% group. Total alkaline protease activity was lower in the SSF groups and higher in the 40% groups (Table 6).

Hepatic glutamate dehydrogenase (GDH) and aspartate aminotransferase (ASAT) activities were not affected by dietary composition. Alanine aminotransferase (ALAT) activity was higher in fish fed the 20 Mix than in the 40 Mix PF diets (Table 7).

Plasma total protein, cholesterol, and triglyceride levels were not affected by the dietary treatments. Glucose levels increased with SSF in the 40% Mix diets (Table 8).

## 4. Discussion

In the present study, the bioprocessing of the PF mixture using SSF induced the production of cellulase and β-glucosidase, detected in the experimental diets. However, xylanase activity detected in the SSFed PF mixture was not detected in the experimental diets, possibly due to their lower thermostability [46] and consequent loss during the diet production and drying process. Fungi carbohydrase production and the associated lignocellulolytic matrix degradation in SSF have been reported for several plant-based ingredients. Soybean and rapeseed and oil cakes submitted to SSF using *R. oryzae*, *A. ibericus*, and *A. niger* reduced acid detergent fiber (ADF) by about 67–69%, and neutral detergent fiber (NDF) by about 53–63% due to the production of carbohydrases [16]. Similarly, the SSF of rice bran with *A. niger* decreased ADF and NDF by 28% and 18%, respectively [47].

Compared to the unfermented diets, the SSFed PF mixture did not affect the total phenolic content but slightly decreased the total dietary antioxidant activity, which could explain the higher mortality registered in the SSFed groups. The fermentation conditions of the SSF of the PF were optimized to maximize the production of carbohydrases [38], so some of the antioxidants may have been lost during the fermentation process. Indeed, depending on substrate composition, SSF can alter the phenolic content composition and the antioxidant activity [48]. SSF of pea-protein-enriched flour with *A. oryzae* or *A. niger* increased total phenolic activity by 31% and 26%, respectively, 6 h post-fermentation [49]. However, SSF of wheat and oat bran with *S. cerevisiae* decreased total phenolic and antioxidant activity after 3 to 4 days of fermentation [50]. The reduction in antioxidant activity during SSF may be due to the production of enzymes, such as laccases and peroxidases [21], which catalyzes the oxidation of antioxidants and phenols [51]. Furthermore, the fermented antioxidant content is also related to the fermentation time. The kinetic production/release of secondary compounds during SSF with *A. ibericus* peaked between the second and third days of fermentation, followed by a decrease, while enzyme activity production started two days after fermentation [38]. 

*A. niger* is classified as among the GRAS fungi by the FDA (US Food and Drug Administration) [52]. In the present study, the mycotoxin contamination of the experimental diets was not detected or, in the case of zearalenone and deoxynivalenol, was below the maximum concentration (2000 and 5000 µg/kg, respectively) established by the European Commission Recommendation (2006/576/EC of 17 August 2006) of deoxynivalenol, zearalenone, ochratoxin A, T-2 and HT-2 and fumonisins present in animal feeds. A residual concentration of ergot alkaloids was detected in the diets, but it is not included in the guidance of the European Commission Recommendation.

SSF with *A. niger* is used to produce commercial enzymes used in aquafeeds, such as Synergen^TM^ [53,54], Allzyme^TM^ [55,56], or Naturgrain^R^ [57], used to improve the digestibility of plant-based diets. In the current study, dietary protein digestibility was generally high, with ADC ranging from 84 to 87%, except for 40 SSF, which decreased to 70%. SSF improved dry matter, protein, and energy digestibility in 20% Mix diets. These results may be due to the reduction in the PF cell walls’ lignocellulosic matrix complexity, production of exogenous carbohydrases and proteases, and reduction in antinutritional factors such as anti-trypsin. The positive effect of ingredient fermentation on diet digestibility has been reported by several authors [30,58,59,60]

The reduction in 40 SSF diet digestibility may be attributed to the presence of high amounts of *A. niger* mycelium and spores, naturally produced during the fungi sporulating life cycle as part of asexual reproduction [61]. These two structures are coated with a chitin–glucan complex [62,63] that is indigested by carnivorous fish, such as European sea bass, due to residual chitinase production [64,65]. Dietary chitin has been shown to reduce feed digestibility in Atlantic salmon (*Salmo salar*) [66], *Nile tilapia* (*Oreochromis niloticus*) [67], and meagre (*Argyrosomus regius*) [68].

In this work, growth performance was unaffected by SSF in the 20% diets, but when fish were fed the 40 SSF diet, growth performance decreased, indicating that high SSF dietary inclusion levels negatively impacted seabass zootechnical performance. Similar results were observed when *Clarias* sp. fingerlings were fed diets including graded amounts of SSF rice bran with *A. niger* as a substitute for soybean meal. *Clarias* sp. growth performance increased when SSF rice bran was included up to 3.75%, decreasing with higher inclusion levels of 5%, 8.75%, and 10% [47]. Nevertheless, SSF of corn husk, rice bran, palm, rapeseed, and soybean meal with a mixture of *Lactobacillus* spp., yeast, and *Bacillus* spp., included at 3% and 5%, increased juvenile gibel carp (*Carassius auratus gibelio* var. CAS V) growth performance [69], while an 8% inclusion had no positive effects. Similarly, the growth performance of Asian sea bass (*Lates calcarifer)* improved when fed diets containing 3%, 8%, and 11% SSF soybean meal with *A. niger* [70]. In turbot (*Scophthalmus maximus* L.), dietary inclusion of a soybean meal fermented with *A. awamori* did not affect growth performance compared to fish fed non-fermented diets [71]. On the contrary, in rainbow trout and whiteleg shrimp (*Litopenaeus vannamei*), the inclusion of 30% soybean and 50% corn gluten fermented with *B. subtilis* U304 decreased growth performance compared to the unfermented control diet [72,73].

As already mentioned, the reduction in growth performance of fish fed the 40 SSF diet may be attributed to decreased diet digestibility. In addition to impairing diet digestibility, the 40 SSF diet also strongly comprised voluntary feed intake. SSF of PF has been reported to improve the palatability of animal feeds and stimulate animal appetite [74]. Nevertheless, Brooks [75] suggested that high concentrations of some metabolites produced by fungi may reduce feed palatability and, consequently, feed intake. 

In this study, the incorporation of the SSF mixture as 20% of the diet increased FE and PER, while the inclusion as 40% decreased both, probably due to the lower growth rate of fish fed the 40 SSF diet. Similarly, rohu fed a diet with *M. oleifera* leaf meal, and Asian sea bass fed soybean meal, both fermented by *A. niger*, showed an increase in FE and PER compared to the unfermented control diet [70,76]. In addition, in juvenile gibel carp and rohu, FE increased in diets containing a 5% feedstuff mixture SSFed with *Lactobacillus* spp., *Bacillus* spp., and yeast [69] or with *S. cerevisiae* [31]. In *Clarias* sp., FE and PER were unaffected by the dietary inclusion of 3.75% SSFed rice bran with *A. niger* but decreased at higher inclusion rates [47]. Similarly, for rainbow trout, a diet incorporating 40% soybean meal or 50% corn gluten, fermented with *B. subtilis* U304, decreased FE [72].

In the present work, the body protein content of juvenile sea bass increased with the 20 SSF diet but decreased with the 40 SSF diet. The increase in protein content was mainly due to the reduced lipid content of the whole body. Similar results were found in rohu juveniles fed diets including SSFed *M. oleifera* leaf meal with *A. niger* [76]. In other species, such as gibel carp [69] and rainbow trout [72], fermented plant-based diets did not affect body macronutrient composition.

During SSF, fungi produce several enzymes, including carbohydrases, amylase, lipase, phytase, laccase, proteases, and others, which remain in the mixture and are incorporated into diets formulated with SSFed ingredients. Dietary supplementation with exogenous enzymes has been reported to influence digestive enzyme activity. For example, lipase and amylase activities increased in Jian carp (*Cyprinus carpio* var. Jian) [77] fed a xylanase-supplemented diet. Similarly, a carbohydrase-supplemented diet also increased the activity of these enzymes in turbot [53] and in white sea bream (*Diplodus sargus*) [57]. However, these enzyme activities were unaffected in rainbow trout juveniles fed a diet supplemented with SSF-produced enzymes [27]. Despite the increased apparent digestibility coefficient of protein observed with the 20 SSF diet, lipase and total alkaline protease activity decreased with the SSFed PF mixture, regardless of the inclusion level. Similarly, the dietary inclusion of 5% SSFed *Ulva rigida* in European sea bass did not affect amylase activity but decreased lipase and protease activities [78]. These results may be attributed to the digestion of PF protein and complex polysaccharides, favoring the access of endogenous enzymes to the substrate. Diets containing 40% PF mixture had higher amylase activity than diets including 20%, probably due to the higher amylaceous ingredients. The decrease in amylase activity in the 40 SSF diet may be due to starch hydrolysis during fermentation. On the contrary, in rohu juveniles, amylase activity increased when fed a diet incorporating 15% or 30% SSFed *M. oleifera* leaf meal [76] or a fermented diet with *S. cerevisiae* [31]. In rainbow trout juveniles, lipase and amylase activity remained unaffected when fed a diet supplemented with SSF-produced enzymes [27].

In general, diet formulation did not affect the hepatic activity of amino acid catabolic enzymes, except for the activity of ASAT, which was higher in 20% than 40% PF group diets. These results may be related to a higher voluntary feed intake and, consequently, protein intake observed in the 20% diet group. Even though some authors consider the amino acid deaminating and transaminating enzymes non-adaptative [79], others have observed a correlation between their activity, particularly GDH, and the quantity/quality of the dietary protein, including in sea bass [80,81].

In the present study, no effect of SSF was observed. Similarly, in Nile Tilapia (*Oreochromis niloticus*) fed with diets containing 10%, 20%, or 30% SSFed wheat protein with *S. cerevisiae*, ALAT and ASAT activities were not affected compared to the non-fermented diet [82]. However, in *Labeo rohita* fed with diets containing 15% or 30% SSFed *M. oleifera* leaf with *A. niger*, ASAT and ALAT activity increased, probably due to reduced dietary antinutritional factors [76].

In this work, glucose increased in the 40 SSF diet, probably due to the increased availability of carbohydrates after fermentation. Plasma glucose levels usually increase when digestible carbohydrates increase in the diet [83]. Similarly, rainbow trout juveniles fed a plant-based diet supplemented with a commercial carbohydrase had higher plasma glucose levels than a control plant-based diet [27]. However, plasma glucose levels decreased when rainbow trout were fed diets containing 30% SSFed soybean and corn gluten meal with *Bacillus subtilis* U304. In European sea bass fed a plant-based diet supplemented with an extract obtained from the SSF of a brewer’s spent grain with *A. ibericus*, the cholesterol, triglycerides, and total protein levels were affected [78]. In large yellow Croaker juveniles (*Larimichthys crocea*), xylanase supplementation did not affect serum glucose, total protein, and cholesterol levels, while triglycerides increased [84]. Moreover, total protein, cholesterol, and triglyceride levels decreased in *L. vannamei* fed graded levels (55, 7.5%, and 10%) of SSFed rapeseed meal and soybean meal with *A. niger*, while serum glucose increased [58,85]. 

## 5. Conclusions

SSF of the PF added cellulase and β-glucosidase activity to the experimental diets, but xylanase was not detected. The dietary total antioxidant level decreased, and phenolic compound activity was not affected by SSF. Mycotoxin contamination was not detected, except for residual levels of zearalenone and deoxynivalenol (100 and 600 times lower than that established by the European Commission Recommendation (2006/576/EC). In diets including 20% PF, SSF did not affect growth but increased protein and energy ADCs, FE, and PER. On the contrary, in diets including 40% PF, SSF decreased growth performance, FE, FI, PER, and diet digestibility. Total alkaline protease activity was reduced via the inclusion of SSFed PF, while the hepatic GDH, ALAT, and ASAT activities, total protein, cholesterol, and triglyceride plasma levels were affected. 

The dietary inclusion of moderate levels of the SSFed PF, up to 20%, improves the overall feed utilization efficiency without negatively impacting European sea bass growth performance.

## Figures and Tables

**Table 1 animals-13-02692-t001:** Ingredient composition and proximate analysis of the experimental diets.

	20 Mix	20 SSF	40 Mix	40 SSF
Feedstuff (% dry weight)				
Fish meal ^1^	17.5	17.5	17.5	17.5
CPSP^® 2^	2.5	2.5	2.5	2.5
Plant-feedstuff mix ^3^	20	0	40	0
SSF plant-feedstuff mix ^4^	0	20	0	40
Wheat gluten meal ^5^	7.5	7.5	7.5	7.5
Corn gluten meal ^6^	13.4	11.4	5.1	1.2
Hemoglobin ^7^	5.0	5.0	5.0	5.0
Wheat meal ^8^	14.2	15.5	4.0	6.6
Fish oil	13.5	14.2	12.9	14.2
Hydrolyzed shrimp ^9^	0.5	0.5	0.5	0.5
Chromium oxide	0.5	0.5	0.5	0.5
Vitamin premix ^10^	1.0	1.0	1.0	1.0
Choline chloride	0.5	0.5	0.5	0.5
Minerals premix ^11^	1.0	1.0	1.0	1.0
Binder ^12^	1.0	1.0	1.0	1.0
Dicalcium phosphate	1.2	1.2	0.3	0.3
Lysine ^13^	0.2	0.2	0.2	0.2
Methionine ^13^	0.2	0.2	0.2	0.2
Taurine ^13^	0.3	0.3	0.3	0.3
Proximate analysis (% dry weight)				
Dry matter (%)	96.8	90.8	96.3	90.4
Crude protein (CP)	42.3	41.4	41.0	41.4
Crude lipid (CL)	17.3	17.4	17.4	17.0
Ash	8.1	9.3	9.4	9.8
Gross energy (MJ/kg)	22.9	22.0	23.4	22.9

^1^ Pesquera Centinela, Steam Dried LT, Chile (CP: 68.45%; CL: 12.56%). Sorgal, S.A. Ovar, Portugal. ^2^ Soluble fish protein concentrate (CP: 80.2%; CL: 15.39%). Sopropeche, France. ^3^ Plant-feedstuff mix (%DM) (CP: 33.0%; CL: 5.5%): 25% rapeseed meal (CP: 40.0%; CL: 5.8%); 25% soybean meal (CP: 51.9%; CL: 3.7%); 25% rice bran (CP: 14.2%; CL: 13.2%); 25% sunflower seed (CP: 40.2%; CL: 2.86%). Sorgal, S.A. Ovar, Portugal. ^4^ SSF plant-feedstuff mix (%DM) (CP: 38.03%; CL: 2.5%): 25% rapeseed meal; 25% soybean meal; 25% rice bran; 25% sunflower seed. Sorgal, S.A. Ovar, Portugal. ^5^ Wheat gluten meal (CP: 80.5%; CL: 1.0%). Sorgal, S.A. Ovar, Portugal. ^6^ Corn gluten meal (CP: 62.0%; CL: 2.8%). Sorgal, S.A. Ovar, Portugal. ^7^ Hemoglobin (CP: 91.5%; CL: 0.4%). Sorgal, S.A. Ovar, Portugal. ^8^ Wheat meal (CP: 14.33%; CL: 2.09%). Sorgal, S.A. Ovar, Portugal. ^9^ Hydrolyzed shrimp (CP: 69.8%; CL: 2.1%). Sorgal, S.A. Ovar, Portugal. ^10^ Vitamin premix (mg kg^−1^ diet): retinol, 18,000 (IU kg^−1^ diet); calciferol, 2000 (IU kg^−1^ diet); alpha tocopherol, 35; menadion sodium bis., 10; thiamin, 15; riboflavin, 25; Ca pantothenate, 50; nicotinic acid, 200; pyridoxine, 5; folic acid, 10; cyanocobalamin, 0.02; biotin, 1.5; ascorbyl monophosphate, 50; inositol, 400. ^11^ Mineral premix (mg kg^−1^ diet): cobalt sulfate, 1.91; copper sulfate, 19.6; iron sulfate, 200; sodium fluoride, 2.21; potassium iodide, 0.78; magnesium oxide, 830; manganese oxide, 26; sodium selenite, 0.66; zinc oxide, 37.5; dicalcium phosphate, 8.02 (g kg^−1^ diet); potassium chloride, 1.15 (g kg^−1^ diet); sodium chloride, 0.4 (g kg^−1^ diet). ^12^ Aquacube, Agil, UK. ^13^ Feed-grade amino acids. Sorgal, S.A. Ovar, Portugal.

**Table 2 animals-13-02692-t002:** Bioactive compound and major mycotoxin analyses of the experimental diets.

	20 Mix	20 SSF	40 Mix	40 SSF
Bioactive compound analysis (% dry weight)				
Cellulase (U/g)	nd	8.1	nd	12.5
Xylanase (U/g)	nd	nd	nd	nd
β-glucosidase (U/g)	nd	3.8	nd	2.3
DPPH (µmol Trolox equivalents/g)	16.6	13.0	17.5	13.2
Total phenols (mg gallic acid equivalents/g)	9.4	8.9	8.6	9.6
Major mycotoxin analysis (µg/kg dry weight)				
Aflatoxin B1	nd	nd	nd	nd
Zearalenone	nd	25.9	nd	8.2
Deoxynivalenol	nd	7.1	nd	nd
T-2 toxin	nd	nd	nd	nd
Fumonisin B1	nd	nd	nd	nd
Ochratoxin A	nd	nd	nd	nd
Sum of Ergot alkaloids	nd	0.1	nd	0.1

nd—non detected.

**Table 3 animals-13-02692-t003:** Growth performance and feed utilization efficiency of European sea bass juveniles fed the experimental diets.

Diets	20 Mix	20 SSF	40 Mix	40 SSF	SEM	Two-Way ANOVA		
						SSF	Inclusion	SSF × Inclusion
Initial body weight (g)	10.0	10.0	10.0	10.0	0.005	ns	ns	ns
Final body weight (g)	25.4 ^B^	23.8 ^B^	21.6 ^Ab^	16.0 ^Aa^	1.1	**	***	*
Weight gain (g kg ABW^−1^ day^−1^)	17.7 ^B^	16.7 ^B^	15.0 ^Ab^	9.3 ^Aa^	1.0	**	***	*
Daily growth index	1.6 ^B^	1.5 ^B^	1.3 ^Ab^	0.7 ^Aa^	0.1	**	***	*
Feed intake (g kg ABW^−1^ day^−1^)	19.1	15.5	17.4	12.2	0.8	***	***	ns
Feed efficiency	0.9 ^Ba^	1.1 ^b^	0.9 ^Ab^	0.8 ^a^	0.04	**	ns	*
Protein efficiency ratio	2.2 ^B^	2.6	2.1 ^Ab^	1.8 ^a^	0.1	**	ns	*
Nitrogen intake (g kg ABW^−1^ day^−1^)	1.5 ^A^	1.3	1.6 ^Ba^	1.8 ^b^	0.1	**	ns	*
Nitrogen retention (g kg ABW^−1^ day^−1^)	0.3	0.4 ^B^	0.3 ^b^	0.2 ^Aa^	0.02	*	***	**
Mortality (%)	0	5	1.7	5	1.0	*	ns	ns

Values are presented as mean (*n* = 3) and pooled standard error of the mean (SEM). Two-way ANOVA: ns—non-significant; (*p* ≤ 0.05); * *p* < 0.05; ** *p* < 0.01; *** *p* < 0.001. If the interaction was significant, SSF and inclusion level were analyzed using a *t*-test; means in the same line with different superscript capital or small letters indicate significant differences (*p* < 0.05) between inclusion levels or SSFs, respectively. ABW: average body weight (initial body weight, IBW + final body weight, FBW)/2. Weight gain: [(Final weight − Initial weight)/Initial weight] × 100. Feed efficiency: wet weight gain/dry feed intake. Daily growth index: ((FBW1/3 − IBW1/3)/time in days) × 100. Protein efficiency ratio: wet weight gain/crude protein intake. Nitrogen intake: (N intake × 1000)/(ABW × time in days). Nitrogen retention: (FBW × FBN) − (IBW × IBN)/(ABW × time in days), where FBN and IBN are final and initial whole-body N content, respectively.

**Table 4 animals-13-02692-t004:** Apparent digestibility coefficients (ADCs) of the experimental diets.

Diets	20 Mix	20 SSF	40 Mix	40 SSF	SEM	Two-Way ANOVA		
						SSF	Inclusion	SSF × Inclusion
ADC (%)								
Dry matter	34.9 ^a^	47.7 ^Bb^	37.6 ^b^	22.6 ^Aa^	3.0	ns	**	**
Protein	85.3 ^a^	87.7 ^Bb^	84.3 ^b^	69.8 ^Aa^	2.3	*	**	**
Lipids	90.9	91.1	91.3	81.6	1.8	ns	ns	ns
Gross energy	62.6 ^a^	68.2 ^Bb^	65.0 ^b^	45.7 ^Aa^	2.9	*	**	**

Values are presented as mean (*n* = 3) and pooled standard error of the mean (SEM). Two-way ANOVA: ns—non-significant; (*p* ≤ 0.05); * *p* < 0.05; ** *p* < 0.01. If the interaction was significant, SSF and inclusion level were analyzed using a *t*-test; means in the same line with different superscript capital or small letters indicate significant differences (*p* < 0.05) between inclusion levels or SSFs, respectively.

**Table 5 animals-13-02692-t005:** Whole-body composition (% fresh weight) of European sea bass juveniles fed with the experimental diets.

Diets	Initial	20 Mix	20 SSF	40 Mix	40 SSF	SEM	Two-Way ANOVA		
							SSF	Inclusion	SSF × Inclusion
Whole-body composition (% fresh weight)									
Dry matter	29.6	32.1 ^Bb^	30.6 ^a^	30.3 ^A^	30.5	0.3	*	*	*
Protein	7.6	17.5	18.1	17.1	16.9	0.3	*	ns	ns
Lipid	16.8	10.6	9.1	9.2	9.2	0.2	ns	ns	ns
Ash	6.5	8.0	8.1	8.7	8.1	0.2	ns	ns	ns
Gross energy (KJ/g)	7.6	8.3	7.5	7.9	7.5	0.1	ns	*	ns

Values are presented as mean (*n* = 3) and pooled standard error of the mean (SEM). Two-way ANOVA: ns—non-significant; (*p* ≤ 0.05); * *p* < 0.05. If the interaction was significant, SSF and inclusion level were analyzed using a *t*-test; means in the same line with different superscript capital or small letters indicate significant differences (*p* < 0.05) between inclusion levels or SSFs, respectively.

**Table 6 animals-13-02692-t006:** Digestive enzyme activity of European sea bass juveniles fed with the experimental diets.

Diets	20 Mix	20 SSF	40 Mix	40 SSF	SEM	Two-Way ANOVA		
						SSF	Inclusion	SSF × Inclusion
Amylase (mU/mg protein)	43.0 ^a^	51.5	67.9 ^Bb^	41.2 ^A^	3.6	ns	ns	*
Lipase (mU/mg protein)	24.9 ^Aa^	42.6 ^B^	38.8 ^b^	43.2	1.9	*	**	*
Total alkaline proteases (mU/mg protein)	310.6	226.5	344.4	330.3	14.7	*	**	ns

Values are presented as mean (*n* = 9) and pooled standard error of the mean (SEM). Digestive enzyme activities are expressed in mU/mg protein. Two-way ANOVA: ns—non-significant; (*p* ≤ 0.05); * *p* < 0.05; ** *p* < 0.01. If the interaction was significant, SSF and inclusion level were analyzed using a *t*-test; means in the same line with different superscript capital or small letters indicate significant differences (*p* < 0.05) between inclusion levels or SSFs, respectively.

**Table 7 animals-13-02692-t007:** Liver intermediary metabolism enzyme activity of European sea bass juveniles fed the experimental diets.

Diets	20 Mix	20 SSF	40 Mix	40 SSF	SEM	Two-Way ANOVA		
						SSF	Inclusion	SSF × Inclusion
GDH (mU/mg protein) ^1^	427.6	435.8	417.1	363.8	23.1	ns	ns	ns
ALAT (mU/mg protein) ^2^	620.9	533.5	484.1	480.5	22.2	ns	*	ns
ASAT (mU/mg protein) ^3^	2425.4	2087.1	1866.4	2070.5	101.4	ns	ns	ns

Values are presented as mean (*n* = 9). Intermediary metabolism enzyme activities are expressed in mU/mg protein. SEM: pooled standard error of the mean. ns—non-significant; (*p* ≤ 0.05); * *p* < 0.05. ^1^ Glutamate dehydrogenase. ^2^ Alanine aminotransferase. ^3^ Aspartate aminotransferase.

**Table 8 animals-13-02692-t008:** Plasma metabolite quantification of European sea bass juveniles fed with the experimental diets.

Diets	20 Mix	20 SSF	40 Mix	40 SSF	SEM	Two-Way ANOVA		
						SSF	Inclusion	SSF × Inclusion
Cholesterol (mg/dL)	89.9	91.0	85.1	103.5	2.9	ns	ns	ns
Total protein (mg/dL)	202.6	197.5	217.7	192.8	192.8	ns	ns	ns
Triglycerides (mg/dL)	753.5	820.5	747.5	732.8	25.6	ns	ns	ns
Glucose (mg/dL)	84.6	79.9	78.9 ^a^	84.5 ^b^	1.9	ns	ns	*

Values are presented as mean (*n* = 9). Plasma metabolites are expressed as mg/dL. SEM: pooled standard error of the mean. ns—non-significant; (*p* ≤ 0.05); * *p* < 0.05. If an interaction was significant, SSF effects were analyzed using one-way ANOVA for 20% and 40% inclusion; means in the same line with different superscript small letters indicate significant differences (*p* < 0.05) between, respectively.

## Data Availability

The data presented in this study are available on request from the corresponding author.

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
