# Peer review of "Solid-State Fermentation as Green Technology to Improve the Use of Plant Feedstuffs as Ingredients in Diets for European Sea Bass (*Dicentrarchus labrax*) Juveniles"

_animals, 2023, doi:10.3390/ani13172692_

Round 1

Reviewer 1 Report

Decision on the manuscript with ID (animals-2508204-peer-review-v1) is “Major Revisions”.

The manuscript by Vieira and coauthors evaluated the potential use of fermented plant stuffs as useful ingredients in diets for juvenile European sea bass. The manuscript is interesting; however, there are revisions that should be considered by the authors when revising their manuscript. The authors should add a continuous line numbering for the revised manuscript in order to help the reviewers with further revisions. I have several questions and the authors should give a point-by-point response to the points below.

Q1. From where the authors got the fungal strain (Aspergillus niger CECT 2088) that used for Solid-State Fermentation? What are the preparation procedures?

Q2. Add appropriate reference for the Solid-State Fermentation process used in the present study?

Q3. How did the authors adjust the spores count in the suspension containing 106 spores per mL? Details should be described.

Q4. Did the sterilization process by autoclaving at 121 °C for 15 minutes did not affect the fungal growth?

Q5. Did the drying procedure at 60 °C not affect the quality of the obtained fermented mixture?

Q6. Why did the authors use the doses at 20% (20 Mix and 20 SSF) and 40% (40 Mix and 40 SSF)?

Q7. a 3 mm thick diameter.

Q8. fish (initial body weight: 70.9 g). The authors should add the value as means ± SD. I think that the initial weights were not similar among all fish.

Q9. fish (initial body weight: 10 g) were formed. Same as the above question.

Q10. Why did the authors feed the fish six days a week and not fed them every day throughout the whole experimental period as a continuous feeding

Q11. killed with anesthesia overdose. Add details and reference.

Q12. (AOAC International, 2016). Use MDPI citation method.

Q13. Table 2: Where are the comments of the authors on the detected levels of mycotoxins in the fermented mixture especially in 20 SSF and 40 SSF? This is an important point. Did these levels not affect the measured parameters in the short run of the feeding experiment? Also, what about the long run effects?

Q14. Table 3: Add a possible attribution for the mortality % obtained among different groups

Q15. Table 6 and Table 7: Add the measuring units for each enzyme.

Q16. Table 8: Add the measuring units for these plasma parameters.

Q17. References: - Authors should follow the MDPI reference style and journal guidelines when revising the references.

Moderate English Editing is required

Author Response

#Reviewer 1

The manuscript by Vieira and coauthors evaluated the potential use of fermented plant stuffs as useful ingredients in diets for juvenile European sea bass. The manuscript is interesting; however, there are revisions that should be considered by the authors when revising their manuscript. The authors should add a continuous line numbering for the revised manuscript in order to help the reviewers with further revisions. I have several questions and the authors should give a point-by-point response to the points below.

Q1. From where the authors got the fungal strain (Aspergillus niger CECT 2088) that used for Solid-State Fermentation? What are the preparation procedures?

R: The authors got the fungal stain Aspergillus niger CECT 2088 from the Spanish collection of type cultivars, Valencia, Spain ( the acronym CECT means - Colección Española de Cultivares Tipo). This information was added to the text.

The preparation procedures are described between lines 106 to 112 "Firstly, the PF mixture was sterilized by autoclave (121 °C for 15 minutes). Then, SSF was performed with 400 g of substrate inoculated with 80 ml of spores suspension containing 106 spores ml-1 and maintained at 25 °C and 75% humidity (w/w) wet basis in tray bioreactors (43×33×7 cm) for 7 days. The substrate was stirred daily to ensure proper substrate oxygenation due to the 5 cm height of the substrate bed, and the tray was covered with plastic film with small holes. After SSF, the fermented mixture was dried at 60 °C for 3 days."

Q2. Add appropriate reference for the Solid-State Fermentation process used in the present study?

R: The reference was added to the text (line 118).

Q3. How did the authors adjust the spores count in the suspension containing 106 spores per mL? Details should be described.

  1. Spores present in a slant of fungi were first suspended with 5ml of spore solution. Then spores were counted and adjusted to 10^6 by counting spores with a Neubauer chamber and adjusting the concentration by dilutions. Additional information was included in the manuscript to accommodate this comment (line 110-112).

Q4. Did the sterilization process by autoclaving at 121 °C for 15 minutes did not affect the fungal growth?

R: The author thanks the Reviewer for the concern. The sterilization was done before the inoculation of the fungus used in the SSF, to make sure that no other microorganisms were presented in the substrate and that the results of the SSF were only attributed to the growth of the fungal stain Aspergillus niger CECT 2088 used in the study. So, in our opinion, autoclaving did not affect Aspergillus niger CECT 2088 growth.

Q5. Did the drying procedure at 60 °C not affect the quality of the obtained fermented mixture?

R: Although the thermostability of the produce enzymes was not determined, most fungal cellulases and xylanases are thermostable at 60°C, and their optimum temperature ranges from 40-60°C or more. Previously, we observed that enzymes produced by SSF of brewer's spent grain, following a similar SSF process described in this study, had a high thermostability (https://www.ncbi.nlm.nih.gov/pubmed/35592554). The drying process is very important, as this fermented mixture was used as a feed ingredient. It is important to ensure that the humidity of the fermented mixture is lower than 10% to avoid mold contamination.

Q6. Why did the authors use the doses at 20% (20 Mix and 20 SSF) and 40% (40 Mix and 40 SSF)?

R: The authors tested two different levels of plant mix, 20% to be more conservative with the plant ingredients inclusion for a carnivorous species such as European seabass and 40 % testing the upper limits of plant ingredients dietary inclusion for this species. Diet 20 Mix was formulated to be similar to a conventional feed, and the 40 Mix diet as a challenging diet.

Q7. a 3 mm thick diameter.

R: Done accordingly.

Q8. fish (initial body weight: 70.9 g). The authors should add the value as means ± SD. I think that the initial weights were not similar among all fish.

R: We thank the Reviewer for the suggestion. The information regarding IBW means and pooled SEM was added in Table 3. We have preferred to use pooled SEM rather than SD because a condition for running a parametric statistical analysis such as ANOVA, as in the present case, is that the variance for all the treatments is the same. Therefore, pooled SEM, i.e., an average estimate for the variance of the means, is given in the table as what we think is the most relevant indicator of the variation of the means to evaluate the importance of the differences between means. The standard deviation for each mean indicates the variation between individual observations behind the means, which we consider not so important in the present context and complicates the "picture" presented by the means. We, therefore, hope that pooled SEMs can be accepted as a measure of variation in the experiment.

Q9. fish (initial body weight: 10 g) were formed. Same as the above question.

R: We thank the Reviewer for the suggestion. The information regarding IBW means and pooled SEM was added in Table 3. However, we did not include the SD in the text for the reasons presented in question 8.

Q10. Why did the authors feed the fish six days a week and not fed them every day throughout the whole experimental period as a continuous feeding

R: We thank the Reviewer for highlighting this topic. However, this is a standard protocol followed in our laboratory as in many other laboratories. Inclusively, there are other published papers where fish were kept unfed during the weekend. Anyway, as all treatments were fed the same way, the feeding pattern would affect all groups similarly.

  1. The reviewer number 2 questioned that fish were unfed for 1.5 days a week during the experimental trial.

Q11. killed with anesthesia overdose. Add details and reference.

R: Done accordingly.

Q12. (AOAC International, 2016). Use MDPI citation method.

R: Done accordingly.

Q13. Table 2: Where are the comments of the authors on the detected levels of mycotoxins in the fermented mixture especially in 20 SSF and 40 SSF? This is an important point. Did these levels not affect the measured parameters in the short run of the feeding experiment? Also, what about the long run effects?

R: We thank the Reviewer for highlighting this important topic. However, the level of recommended maximum limits for mycotoxins in feed (100–500 µg/kg for ZEA and 900–5000 µg/kg for DON), are so far from the levels detected in the experimental diets that, despite discussed, did not deserve further attention by the authors (25.9 and 8.2 µg/kg for ZEA and 7.1 µg/kg).

Q14. Table 3: Add a possible attribution for the mortality % obtained among different groups

R: Done accordingly. Additional information was added in lines 365-366.

Q15. Table 6 and Table 7: Add the measuring units for each enzyme.

R: Done accordingly.

Q16. Table 8: Add the measuring units for these plasma parameters.

R: Done accordingly.

Q17. References: - Authors should follow the MDPI reference style and journal guidelines when revising the references.

R: Done accordingly. 

Reviewer 2 Report

This manuscript describes a trial with juveniles European sea bass fed either with a plant protein mixture or with the same mixture fermented by Aspergillum niger. The content of the manuscript may be of interest, a lot of job done has been reported, nevertheless minor as well as major problems exist. Simple summary: some mention about the results obtained with the 40% SSF as well, would be appreciate. Abstract: the  trial  length is not reported. Introduction: Listing the protein sources utilized for fish feed, the authors limited the description to fish meal and plant feedstuffs (PF), while other sources, as PAT, insects, SCPs should be mentioned as well. Mentioning the Food and Drug Administration, the acronym was GRAS and not FDA? Material and Methods: Among the experimental diets, a real Blank with a conventional feed was not included, as only non fermented (Mix) and fermented PF (SSF) were compared. On table 1, the Soluble Fish Protein Concentrate are reported as CPSP, while SFPC would be more comprehensive. Describing the digestible trial, the authors missed to indicate the fish species, they also missed to report the utilized marker that may be assumed only by the formula (1) as Cr2O3, nevertheless its concentration is missed as well. Describing the Growth trial, it is not clear if during the 2 months acclimatization the fish were still fed with the previous commercial diet. A MAJOR problem occurs when the length of the trial is reported, as 50 days is definitively a too short time for such kind of study, also by considering that during the trial, the fish little more than double their biomass, only. This may be a good reason to REJECT the manuscript. The authors should explain why the trial was stopped at that point and convincingly explain a good reason for that. By reporting the water quality during the trial, the author should indicate also the fluctuations of the reported parameters, as well as the concentration of the nitrogenous compounds. Results: The authors declared that an intermediate monitoring of the fish performances were done after 4 weeks, nevertheless the data were not reported. On table 2 evidence exists that the concentrations of Zeralenone as well as of Deoxynivalenol where higher in 20 SSF than in 40 SSF, while it would sound more reasonable viceversa. On table 3, mortality was definitively higher among SSFs  (5%) than among Mix (0-1.7%): one may wonder what it would happened in a longer trial! Also, it's hard to believe that no sign of stress where observed, as the authors affirm. Other data, as final body weight, by assuming that all the initial weight were the same (not clear), show that even the diet SSF 20 produced lower growth than 20 Mix. Doubts also arise among the other performances, even if we just forgot the SSF 40 column.

In conclusion, the only reason this Reviewer doesn't suggest immediately to reject this manuscript, is because the authors have done a huge amount of job. Nevertheless they should convincingly give a good reason for a trial of only 50 days. 

Only little improvements needed

Author Response

#Reviewer 2

This manuscript describes a trial with juveniles European sea bass fed either with a plant protein mixture or with the same mixture fermented by Aspergillum niger. The content of the manuscript may be of interest, a lot of job done has been reported, nevertheless minor as well as major problems exist. Simple summary: some mention about the results obtained with the 40% SSF as well, would be appreciate.

R: Done accordingly.

Abstract: the  trial  length is not reported.

R: Done accordingly.

Introduction: Listing the protein sources utilized for fish feed, the authors limited the description to fish meal and plant feedstuffs (PF), while other sources, as PAT, insects, SCPs should be mentioned as well.

R: Done accordingly.

Mentioning the Food and Drug Administration, the acronym was GRAS and not FDA?

R: Done accordingly.

Material and Methods: Among the experimental diets, a real Blank with a conventional feed was not included, as only non fermented (Mix) and fermented PF (SSF) were compared.

R: The authors thank the Reviewer for the suggestion but diet 20 Mix was formulated to be similar to a conventional feed.

On table 1, the Soluble Fish Protein Concentrate are reported as CPSP, while SFPC would be more comprehensive.

R: CPSP® is an acronimosn than menas Concentrés de Protéines Solubles de Poisson, as it is produced in France. The symbol ® was added to the table.

Describing the digestible trial, the authors missed to indicate the fish species, they also missed to report the utilized marker that may be assumed only by the formula (1) as Cr2O3, nevertheless its concentration is missed as well.

R: The text was modified to accommodate the Reviewer's concerns.

Describing the Growth trial, it is not clear if during the 2 months acclimatization the fish were still fed with the previous commercial diet.

R: The text was modified to accommodate the Reviewer's concerns.

A MAJOR problem occurs when the length of the trial is reported, as 50 days is definitively a too short time for such kind of study, also by considering that during the trial, the fish little more than double their biomass, only. This may be a good reason to REJECT the manuscript. The authors should explain why the trial was stopped at that point and convincingly explain a good reason for that.

R: We thank the Reviewer for the suggestion. The fish fed the conventional feed, the diet 20 Mix, almost triplicate the initial size, revealing a growth performance within the expected for this fish size. The scientific community considers that the fish should at least double the initial size to assess the effect of different experimental diets. According to the Guidance of Additives and Products or Substances Used in Animal Feed (FEEDAP) on the assessment of the safety of feed "The necessary minimum duration of tolerance trials is dependent on the animal species/category and defined the minimum for growing fin fish as 90 days or until initial body weight is double. For more details, please visit the annex on page 12, also available at  https://www.efsa.europa.eu/sites/default/files/consultation/170406.pdf

By reporting the water quality during the trial, the author should indicate also the fluctuations of the reported parameters, as well as the concentration of the nitrogenous compounds.

R: The text was modified to accommodate the Reviewer's concerns.

Results: The authors declared that an intermediate monitoring of the fish performances were done after 4 weeks, nevertheless the data were not reported.

R: The main objective of this trial was to assess the growth and feed utilization performance of the European seabass after the fish duplicated their size. This intermediary weighing aimed to determine the date of the final weighing. Therefore, the authors did not include that information in the manuscript. The text was also modified to accommodate the Reviewer's concerns.

On table 2 evidence exists that the concentrations of Zeralenone as well as of Deoxynivalenol where higher in 20 SSF than in 40 SSF, while it would sound more reasonable viceversa.

R: We agree with the Reviewer's commentary and expected that as well. Nevertheless, in Table 2 we presented the results obtained in the mycotoxins analysis, and to our surprise were higher in the 20 SSF diet. These results are probably due to the very low levels of both mycotoxins close to the detection of the methods used. However, this is only a speculation, so we decided not to include this in the manuscript.

On table 3, mortality was definitively higher among SSFs  (5%) than among Mix (0-1.7%): one may wonder what it would happened in a longer trial! Also, it's hard to believe that no sign of stress where observed, as the authors affirm. Other data, as final body weight, by assuming that all the initial weight were the same (not clear), show that even the diet SSF 20 produced lower growth than 20 Mix. Doubts also arise among the other performances, even if we just forgot the SSF 40 column.

R: We agree with the Reviewer's opinion. Plasma glucose concentration is a parameter used to assess fish stress status and our results indicate that only fish the diet 40 SSF would possibly be more stressed compared to fish fed the diet 40 mix. So the sentence "No signs of stress and disease were observed during the growth trial. Mortality was low but higher in the SSF groups than in the non-SSF groups (Table 4)." was modified to "No signs of disease were observed during the growth trial. Mortality was low but higher in the SSF groups than in the non-SSF groups (Table 4)."

In conclusion, the only reason this Reviewer doesn't suggest immediately to reject this manuscript, is because the authors have done a huge amount of job. Nevertheless they should convincingly give a good reason for a trial of only 50 days.

R: As we stated before, the fish increased the initial body weight 2.5 times with the conventional diet (20MIX), and in our opinion, this growth is sufficient to assess the dietary effects in growth and feed utilization. Our opinion is fundaments based on the EFSA recommendation. According to the EFSA Guidance of Additives and Products or Substances Used in Animal Feed (FEEDAP) on the assessment of the safety of feed determined that the "The necessary minimum duration of tolerance trials is dependent on the animal species/category and defined the minimum for growing fin fish as 90 days or until initial body weight is double. For more details, please visit the annex on page 12, also available at  https://www.efsa.europa.eu/sites/default/files/consultation/170406.pdf.

Round 2

Reviewer 1 Report

The authors properly addressed the comments raised by the reviewer.

Minor comments

Reviewer 2 Report

In this corrected manuscript, the authors responded to many of the points underlined by this Referee. Nevertheless, some major questions did not receive any answer. For instance, in the first version it was reported that: "The trial lasted for 50 days, and an intermediate weighting was carried out four weeks after the start of the trial." The authors were  asked to report the intermediate weights at 4 weeks, but instead to respond, at the line 172 they've just removed the sentence!!! The authors were requested to explain why the experiment lasted after 50 days only, instead the canonic 12 weeks, in alternative to triplicate their weight. In this case fish not only missed to triplicate their biomass but hardly duplicate their original weight. No responses were provided by the authors, related this precise and important referee's comment. A comment  was related to the fact that a control group, for instance fed with the commercial feed utilized for acclimatization, was not included. No answers! At line 402, the explications of the reduced nitrogen retention with the diet 40SSF is not convincing, as long as with 20SSF the nitrogen retention is higher than with both the diets Mix. Because of all such not resolved major points, the opinion of this Referee is that the manuscript should be rejected, not joining the standard for publication on Animals.

English acceptable. Just minor improvements after a mother tongue lecture may be requested